# Modeling the iron storage protein ferritin reveals how residual ferrihydrite iron determines initial ferritin iron sequestration kinetics

Joseph Masison[1], Pedro Mendes[1,2]*

1 Center for Cell Analysis and Modeling, University of Connecticut School of Medicine, Farmington, CT, United States of America, 2 Department of Cell Biology, University of Connecticut School of Medicine, Farmington, CT, United States of America

* pmendes@uchc.edu

**Data Availability Statement:** All relevant data are within the manuscript and its Supporting Information files. The models are in the BioModels

## Abstract

Computational models can be created more efficiently by composing them from smaller, well-defined sub-models that represent specific cellular structures that appear often in different contexts. Cellular iron metabolism is a prime example of this as multiple cell types tend to rely on a similar set of components (proteins and regulatory mechanisms) to ensure iron balance. One recurrent component, ferritin, is the primary iron storage protein in mammalian cells and is necessary for cellular iron homeostasis. Its ability to sequester iron protects cells from rising concentrations of ferrous iron limiting oxidative cell damage. The focus of the present work is establishing a model that tractably represents the ferritin iron sequestration kinetics such that it can be incorporated into larger cell models, in addition to contributing to the understanding of general ferritin iron sequestration dynamics within cells. The model's parameter values were determined from published kinetic and binding experiments and the model was validated against independent data not used in its construction. Simulation results indicate that FT concentration is the most impactful on overall sequestration dynamics, while the FT iron saturation (number of iron atoms sequestered per FT cage) fine tunes the initial rates. Finally, because this model has a small number of reactions and species, was built to represent important details of FT kinetics, and has flexibility to include subtle changes in subunit composition, we propose it to be used as a building block in a variety of specific cell type models of iron metabolism.

## Introduction

Computational modeling is becoming ubiquitous in biology. Because many components of biological systems appear in multiple systems, it is common that parts of a new model had already been developed for previous ones. Thus, creating models designed to be easily extensible facilitates creation of new and more complex models. Model reuse reduces the need for repeating literature searches for kinetic parameters, initial conditions and modeling

Database as MODEL2211030001 and
MODEL221030002.

**Funding:** P. M. is funded by the NIH grant R24
GM137787, National Resource for Mechanistic
Modeling of Cellular Systems. The funders had no
role in study design, data collection and analysis,
decision to publish, or preparation of the
manuscript.

**Competing interests:** The authors have declared
that no competing interests exist.

specifications [1] and improves reproducibility [2]. Construction of reusable models is greatly assisted by adherence to model standards (file formats, software, etc.) including the Systems Biology Markup Language (SBML) [3], and increased and standardized annotation [4, 5] including Systems Biology Graphical Notation (SBGN) [6] and ODD protocol [7, 8]. Repositories of models and model parts, such as BioModels [9–13] and ModelBricks [1] are also crucial resources to assist with model extension. The innovation of ModelBricks reinforces the concept that some models should be built with the explicit purpose of serving as components for future larger models. Here we contribute to the pool of extensible models by developing a model component that is required in any cellular model focused on iron metabolism, namely the sequestration of labile iron by ferritin.

Iron is an essential metal for life that is dangerous in excess because it can promote oxidative stress. As a result, animals, and particularly mammals, have mechanisms for limiting both iron deficiency and overload. Failure to limit either causes pathology. At the cellular level, the cytoplasmic pool of soluble $Fe^{2+}$ and other forms of weakly-bound iron–known as the "labile" iron pool (LIP)–must be kept small as it is capable of promoting oxidative damage [14]. A necessary component to keep low levels of LIP is the storage protein ferritin (FT) [15–17]. Ferritin forms a roughly spherical, hollow protein cage made up of 24 subunits that can sequester up to around 4300 iron atoms per cage [16, 18]. Because of FT's universal involvement in iron regulation [15], models of iron metabolism generally must include FT making an extensible submodel of FT particularly useful. Such a model is described in detail in this publication.

To store iron, FT imports and converts soluble ferrous iron ($Fe^{+2}$) from the LIP into an insoluble ferric iron ($Fe^{+3}$) mineral core [19–21]. The storage process begins as a ferrous ion moves into the FT cavity through one of 8 existing pores [22]. After entry, the ferrous iron participates in a series of reactions resulting in its oxidation to the ferric state [23, 24]. Once in the ferric form, it is effectively trapped within the cage, requiring reduction before it can be released [25]. The final steps of iron sequestration, denoted mineralization, convert the ferric iron intermediates to mineral ferrihydrite [21, 26]. Mineralization is catalyzed by the mineral core itself, promoting incorporation of ferric iron intermediates thereby growing the existing mineral. FT has 2 subunit types, heavy (H) and light (L) that impact this iron storage process [27, 28] and are expressed in a tissue dependent way. In humans, the liver has been shown to contain L-enriched FT and the heart, H-rich FT. The H subunit has a ferroxidase active site capable of catalyzing the oxidation of ferrous iron [29]. The ferroxidase accepts 2 ferrous irons and releases into the FT cavity a diferric peroxo complex ferric intermediate (DFP) [21, 30]. Though iron sequestration and the steps leading to DFP formation are well described in the literature, it is unclear how the ferric DFP is transformed into a ferrihydrite mineral. Overall, there is still a limited quantitative understanding of iron sequestration by ferritin.

Other models of ferritin iron storage dynamics have been previously reported [31–33], of which two are especially noteworthy. The first, by Macara *et al.* 1972 [31], was built to test the feasibility of a hypothesis that the iron previously incorporated into a FT cage changes the rate of incorporation of subsequent iron. Its phenomenological nature is explanatory, but the lack of detail and use of relative quantities limits its potential to be included in larger models. The second one, by Salgado *et al.* 2010 [32], is based on a detailed mechanism and tracks sequestration of packages of 50 iron atoms. This model was developed to better establish the mechanism of iron sequestration by FT but it also contains issues that undermine its utility. Most significantly, the model considers a maximum of 2500 atoms within FT (which can hold a maximum of 4300) and the effect the mineral core has on FT iron sequestration is not considered. Additionally, this model contains a large number of reactions, which makes it too complex for inclusion as a part of a larger model. Both the Macara et al. and Salgado et al. models have been useful in helping us understand FT iron sequestration dynamics, but neither have been

published electronically in suitable standard formats, which also deters their re-use. Our present objective is thus to create a model in a suitable standard format, with an appropriate level of detail that makes it useful for incorporation into larger cellular models, but which still captures essential aspects of the dynamics of FT iron sequestration. Beyond this purpose of making the model suitable for inclusion in larger models, it is also useful in itself to allow exploration of how FT characteristics influence its iron sequestration behavior. Specifically the model addresses 1) how the iron present within a FT cage affects its iron sequestration dynamics 2) how *in vitro* experimentally derived kinetic constants related to FT iron sequestration manifest in the behavior of live cells, 3) the impact of FT subunit composition, and 4) the dynamics of iron release from FT.

## Methods

### Software

All simulations were carried out with COPASI version 4.35–4.36 [34–36] on a Windows 10 computer with an Intel(R) Core(TM) i7-4770 CPU @ 3.40GHz. Some COPASI simulations were driven by Python scripts, (version 3.9.7) using the library BasiCO, a Python API for COPASI [34–36]. Tasks were either run directly using the COPASI graphical user interface and collecting their output into.tsv files, or the task was run using the BasiCO scripts. Plots were created using Python's Matplotlib library, (version 3.4.3), and exported to vector graphics SVG files and later refined using Inkscape (https://inkscape.org/). Experimental data were gathered from the supplemental materials of the relevant publications, or digitized from their figures using an image digitizer (https://automeris.io/WebPlotDigitizer/).

### Model definition

The model is composed of a set of ordinary differential equations that approximates the process of cytoplasmic ferrous iron FT internalization and subsequent mineralization into a ferrihydrite core (as detailed in the Introduction). It explicitly tracks the concentration of "free" ferrous iron in solution (LIP, labile iron pool), ferric iron intermediate (DFP), mineralized iron (inside FT), and the total concentration of FT polymer cages. The model design is motivated to satisfy two somewhat opposing characteristics: 1) contain sufficient mechanistic detail to allow representation of important properties, and 2) be as minimal as possible for easy inclusion in larger models. To satisfy both these characteristics the well described, but complex process of FT iron sequestration was simplified into three steps; oxidation, nucleation, and mineralization as illustrated in **Fig 1**. These three steps provide a modest level of mechanistic detail, while also capturing the process in its entirety. This simplification allows for tracking the amount of iron that enters FT without having to follow each individual atom. These three steps correspond to the experimental methods by which the FT iron sequestration process has been studied, reflected by the availability and type of data surrounding each step. Several experimental studies tracked the formation of DFP (oxidation) [19, 20, 37, 38] and the mineral core (nucleation/mineralization) [21, 39, 40] from an initial iron dose via light absorbance at 650 nm and 305 nm wavelengths respectively. In fact, experimental data has prompted proposal of similar reaction schemes providing further motivation for our simplification [41, 42].

The model considers four chemical species: 1. **LIP**, the labile iron pool (soluble or easy to solubilize ferrous iron in cytoplasm, often referred to as "free" iron), 2. **DFP,** the oxidation intermediate diferric peroxo complex, 3. **core,** the iron that is part of the ferrihydrite mineralized core, and 4. **FT,** the ferritin 24-subunit polymers. There are four reactions in the model; 1. **Oxidation** converts two LIP to DFP, 2. **Reduction** converts DFP back to two LIP, 3. **Nucleation** converts two DFP to form a new core crystal (core), and 4. **Mineralization** adds one

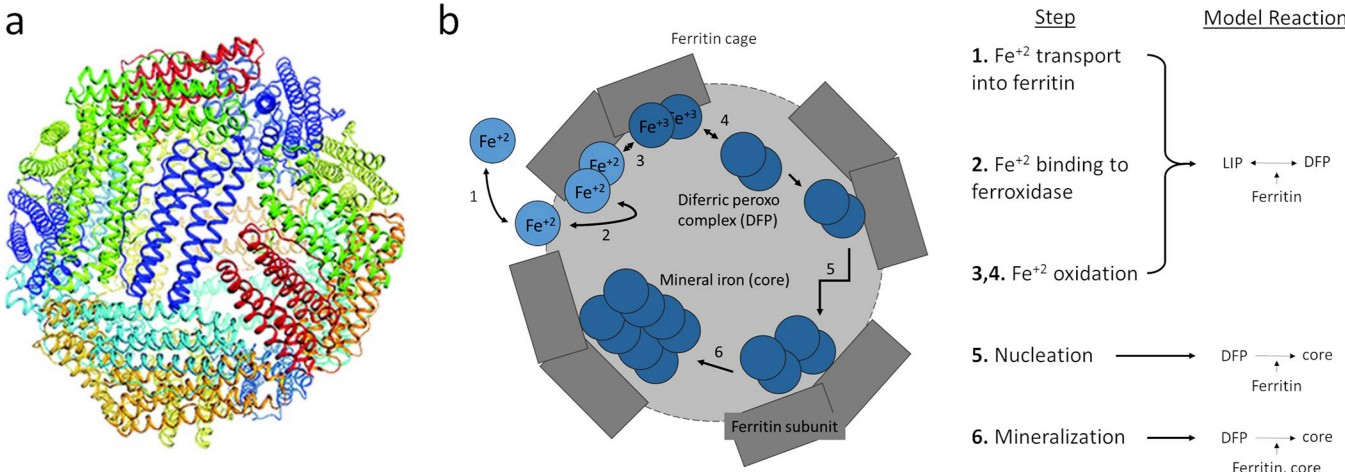

**Fig 1. Model overview. a.** Structure of a 24 subunit ferritin polymer cage, (reproduced from Ebrahimi et al. [43] without modification). **b.** Cartoon representing a single 24-subunit polymer FT cage illustrating the iron sequestration process broken up into a series of steps (as if the cage was cut in half and the dark grey rectangles signify the FT subunits with gaps for LIP transport into the cage cavity and the light grey representing the internal cavity that fills up with iron). The model represents the process of ferritin iron sequestration by approximation of the steps into four reactions, the first two of which are combined in the figure because they compose a reversible reaction. **Oxidation** represents the combined processes of ferrous (+2 charge, light blue) iron transport into ferritin through the three-fold pore, binding to the ferroxidase center located on H subunits, and the oxidation to the ferric state (+3 charge, dark blue, in the form of DFP), and unbinding from the ferroxidase (steps 1,2,3,4 respectively), with **Reduction** the reverse. **Mineralization** is the process of incorporating the two iron ions that are part of DFP into the mineral core (step 6). The rate of this process depends on the amount of iron already in the core. It proceeds as DFP that has unbound (step 4) from the ferroxidase reacts with the preexisting core and is incorporated into it. **Nucleation** represents a special case of mineralization, one where there is no preexisting core so it must be formed from several existing DFP molecules (step 5).

DFP to an existing growing crystal. Note that while the ferrihydrite that forms the core is a solid, the model species "core" accounts for the total iron in all core crystals (of all FT cages), and its value represents the concentration of iron that would be released if all the ferrihydrite crystals would be solubilized into the cytoplasm. The reactions and their kinetic properties are summarized in **Table 1** and a rationale for the determination of their rate laws is outlined below.

**Table 1. Model mathematical representation.** Shown below are the four reactions in the model, their stoichiometries, rate laws, and rate laws parameter values. The first two reactions, oxidation and reduction are parameterized from published data. Nucleation and mineralization were parameterized using parameter estimation.

| Reaction Name | Reaction | Rate Law | Parameter | Parameter Value | Ref. |
|---|---|---|---|---|---|
| **Oxidation** | 2 LIP → DFP | $\frac{k_{cat} \times \frac{H+rO}{24+rO} \times FT \times LIP^n}{Km^n + LIP^n}$ | $k_{cat}$ | 591 s$^{-1}$ | [20] |
| | | | $Km$ | 0.35 mM | [38] |
| | FT-cage | | $n$ | 1.3 | [38] |
| | | | $rO$ | 2 | - |
| **Reduction** | DFP→2 LIP | $k_{deg} \times DFP$ | $k_{deg}$ | 0.2605 s$^{-1}$ | [20, 38] |
| **Nucleation** | 2 DFP→4 core | $k_{cat} \times DFP^2 \times FT \times \frac{L+rN}{24+rN} \times \frac{Ki^n}{Ki^n + core^n}$ | $k_{cat}$ | 5x10$^7$ s$^{-1}$ | - |
| | | | $Ki$ | 0.4615 mM | |
| | FT-cage, core | | $n$ | 4 | |
| | | | $rN$ | 50 | |
| **Mineralization** | DFP→2 core | $\frac{k_{cat} \times DFP \times core}{Km + DFP} \times \frac{Ki^n}{Ki^n + core^n} \times \frac{4300^m - apc^m}{4300^m}$ | $k_{cat}$ | 0.101564 s$^{-1}$ | - |
| | | | $Km$ | 5x10$^{-6}$ M | |
| | core | | $Ki$ | 4.6458 mM | |
| | | | $n$ | 4 | |
| | | | $m$ | 8 | |

## Reaction one: Oxidation

$$\frac{kcat \times \frac{H+rO}{24+rO} \times FT \times LIP^n}{Km^n + LIP^n} \qquad (1)$$

The oxidation of ferrous iron (LIP) to DFP is governed by Hill kinetics [19]. Accordingly, in the model the **Oxidation** rate law contains the parameters $k_{cat}$ (the catalytic turnover number), $K_m$ (the Michaelis constant), and $n$ (the Hill coefficient) and depends on the concentration of FT and LIP. Since a FT cage is made of 24 subunits of 2 distinct types (H and L) and only H subunits have a ferroxidase active site, FT cages with a different number of H subunits have different oxidation rates. This variation necessitates addition of two additional parameters to the rate law, 1) $H$, the number of H subunits (a number between 0 and 24) and 2) $rO$, a scaling factor representing the efficiency of oxidation of an L-homopolymer. The $H$ parameter is the number of H subunits per cage. The $rO$ parameter needs to be included because despite the L subunit containing no known ferroxidase, L homopolymers still appears to catalyze the formation of ferric iron within FT, though with more than a one fourth reduction in rate [44]. To include this phenomenon, $rO$ is set so that when $H$ equals 0 this L driven oxidation can be included (in other words, oxidation rate is not 0 with 24 L subunits). Because there are limited data on how oxidation occurs in the absence of an H subunit and on an appropriate $rO$ value, the value of $rO$ is set empirically at 2. The value of 2 results in the rate of oxidation of a FT cage with only $L$ subunits being 10-fold less than one with only $H$ subunits.

$$kdeg \times DFP \qquad (2)$$

## Reaction two: Reduction

Reduction of DFP back to ferrous iron follows mass action kinetics [19]. The reduction reaction rate is dependent on the concentration of DFP and the rate constant, $k_{deg}$. The validity of this modeling choice is increased by measurement of the value for this rate constant by an additional independent source [20].

### Reaction three: Nucleation

$$kcat \times DFP^2 \times FT \times \frac{L+rN}{24+rN} \times \frac{Ki^n}{Ki^n + core^n} \qquad (3)$$

Nucleation represents the formation of a new crystal within FT, where 2 DFP molecules come together to form the nucleus of a new ferrihydrite crystal (core). A nucleation event producing a new crystal may occur whether a cage has a crystal already present or not. However, it appears that most FT iron cores are the result of relatively few nucleation events [24, 45, 46]. While the function and role of L subunits are still under investigation [45, 47], this subunit is thought to aid in the nucleation process [47, 48]. Because little is known about the mechanism of nucleation we developed an empirical rate law for this reaction. It is based on mass action, where the rate is proportional to the square of the concentration of DFP (since two of these molecules react), and the concentration of FT cages. Then, to include the different effects of the L and H subunits, the concentration of FT is multiplied by a factor that accounts for their proportion, in a manner similar to the $H$ and $rO$ parameters of the oxidation rate law, in this case the parameters are named $L$ and $rN$. Since the degree to which L is superior in catalyzing nucleation compared to H subunits is not as well described, $rN$ has a value such that polymers of vastly different L ratios do not have variations in nucleation rate as quantitatively large as

the range in oxidation rates of ones of varying H subunits. Lastly, the rate law includes a term for product inhibition which accounts for the decreasing probability of new crystal formation with increasing size of the already existing core, which is characterized by an inhibition constant $Ki$ and a Hill coefficient $n$ (to allow for cooperativity of this effect).

## Reaction four: Mineralization

$$\frac{kcat \times DFP \times core}{Km + DFP} \times \frac{Ki^n}{Ki^n + core^n} \times \frac{4300^m - apc^m}{4300^m} \tag{4}$$

The mechanism of mineralization is also not fully understood yet and thus is also described here by an empirical rate law, composed of three terms representing the catalytic effect of core itself, the inhibition of the reaction at high substrate (core) concentration, and the known (empirical) limit of 4300 iron atoms per core. Since the growth of the core likely happens at the inner surface of the ferritin cage, the kinetics should be hyperbolic in the concentration of the substrate DFP (following Langmuir adsorption kinetics [49]), with the amount of existing core taking the role of the catalyst (and product). Data from Harrison et al., 1974 [40] show that this rate peaks around 1500–2000 iron atoms per core (APC), with lower rates at higher APC values. Thus the rate is both limited by *core* availability, when the value of *core* is low, but also inhibited when the value of *core* is high. This effect is modeled by a term that reduces the rate as the core increases (an uncompetitive inhibition term). Lastly, a term is included to make the rate tend to zero as the value of APC approaches 4300, the maximum observed number of iron atoms per core.

The full set of equations forming the model is included in the Supporting information.

### Calibration

For the oxidation and reduction reactions, the rate laws have been explicitly described and measured in previous literature [19, 20, 32, 37, 38], so their values were adopted here (**Table 1**). For the mineralization and nucleation rate laws, the parameter values were determined by nonlinear least-squares using the parameter estimation task in COPASI using a combination of optimization algorithms (Genetic algorithm, Hooke and Jeeves, and Particle swarm). Published experimental data on initial rates of mineral formation as a function of the number of initial iron atoms per core (APC) [40] were used for the fitting. The range of parameters estimates for rate constants were constrained by knowledge of the chemical properties of mineralization compared to the oxidation step ($K_m$ value is near DFP concentration, $k_{cat}$ is smaller than oxidation $k_{cat}$, etc.). The result of the estimation enabled a reasonable fit to the experimental data, shown in Fig 2A.

### Validation

Model validation is an important step which is designed to test the model in conditions different than reflected in the fitting data. Before determining the numerical values of the model parameters, datasets from three studies that measured FT iron kinetics under conditions different than those used for calibration were set aside *a priori*. To validate the model its output was compared to these datasets. The first one measured DFP formation rate as a function of initial iron concentration [38], the second measured mineralization over time following one or several additions of ferrous iron [21], and the third measured mineralization over time with different initial ferrous iron concentrations [39]. These experimental datasets all came from different laboratories, using different setups, and using different iron loading conditions.

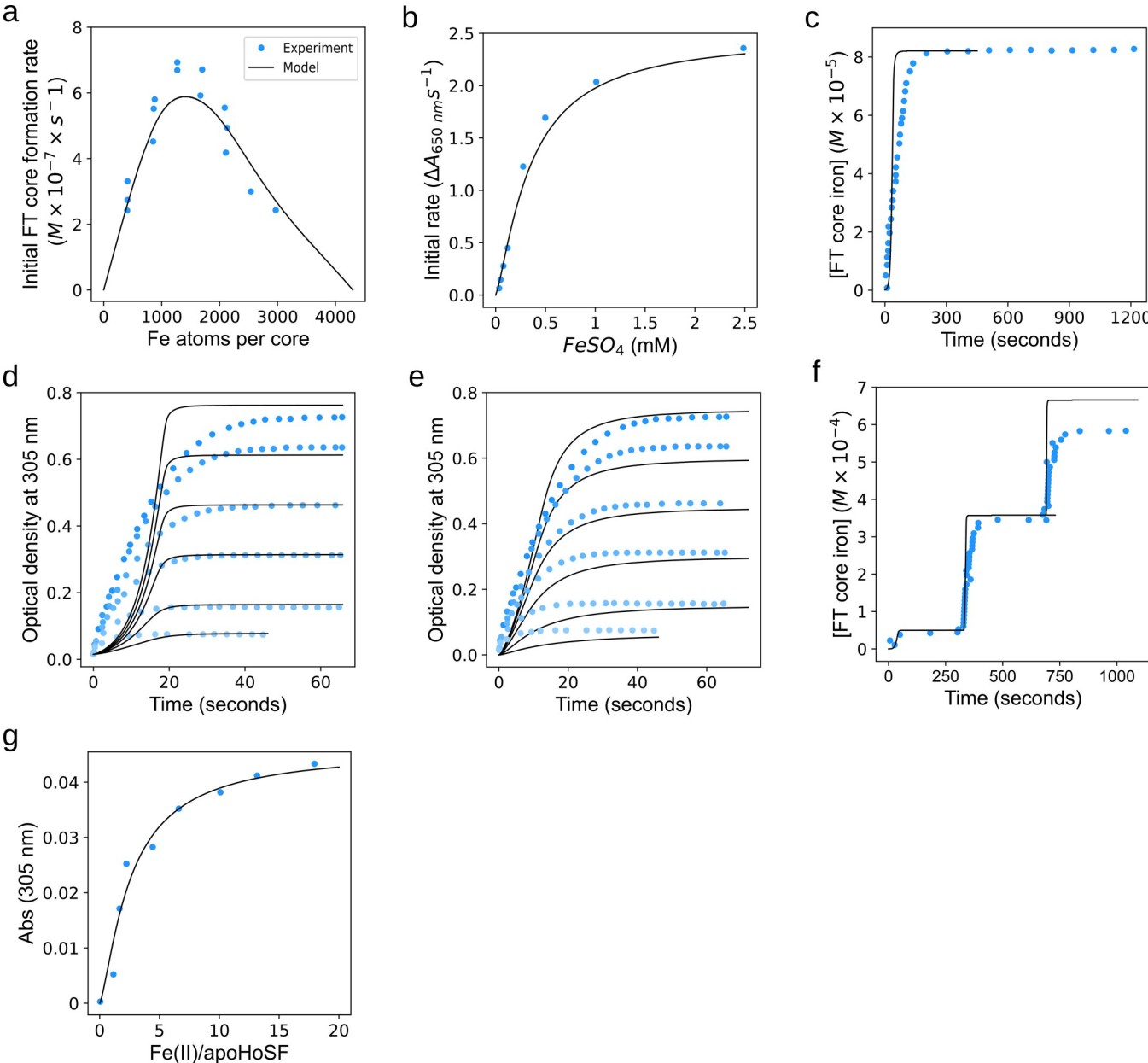

**Fig 2. Model calibration and validation. a. Calibration.** Using data from Harrison 1974 [40], parameter estimation was performed to determine the parameters for mineralization and nucleation. Following the parameter estimation, the model produced output (black line) that was a good representation of the experimental data (blue circles). Conditions: 4 H-chains/FT, 2.32 μM FT, 470 Fe atoms per FT cage **b-g. Validation.** To validate the model, the model was run against four independent experimental conditions. For each experiment, the model parameters were preserved, only the initial concentrations were altered to match the experimental conditions. **b.** DFP formation over time (from Tosha 2008 [38] Fig 2B, Conditions: 24 H-chains/FT, 4.2 μM FT, 1–600 Fe atoms per FT cage), **c.** Iron addition and mineralization (from Zhao 2003 [21] Fig 2, Conditions: 4 H-chains/FT, 0.2 μM FT, 410 Fe atoms per FT cage). **d** and **e.** Mineralization from varied initial iron concentrations (from Fig 1 of Bou 2019 [39]), Conditions: 24 H-chains/FT, 0.5 μM FT. Each time course represents an experimental condition where the FT concentration is the same and the initial iron added was varied. From bottom to top, the amount of iron atoms added per FT cage was 42, 100, 200, 300, 400, 500. **f.** Iron addition and mineralization (from Zhao 2003 [21] Fig 3, Conditions: 24 H-chains/FT, 0.5 μM FT, 100 and then 700 and 700 Fe atoms per FT cage). **g.** Iron addition and mineralization (from Zhao 2005 [41] Fig 2B, Conditions: 4 H-chains/FT, 22 μM FT, 0–20 Fe atoms per FT cage).

Validation revealed that the model parametrization carried out above is satisfactory in that it reproduces the validation data appropriately (Fig 2B–2G). Validation results indicate that the model displays a stronger cooperative effect on DFP mineralization rate than was observed experimentally. In Fig 2D, the modeled mineralization curve is initially lower than the experimental observations, while the model also reaches a steady-state core concentration faster than in the experiments. This difference may not be too significant because in the cellular context, which is the target context for this model, the important timescale is in minutes or greater, while the differences in Fig 2D are in the first 20 seconds. The model behaves more closely to the data in Fig 2D if parameter values change within +/- 10%, as depicted in Fig 2E, though such parameter values would then fit Fig 2B and 2C more poorly. For the final version of the model we adopted the parameter values of Fig 2A–2D and 2f with the knowledge that if required by any future use, a small change in those values could bring the data closer to the data of the Bou et al. 2019 experiment [39]. The results show the model can perform well in predicting FT behavior at cellular timescales, but if the precise tracking of early mineralization events is to be captured, a model with increased detail—but possibly less suited to a cellular model—should be created. Overall we believe these validation results indicate that our model is useful to represent the action of ferritin on iron sequestration as a building block in cellular models.

## Results

Having obtained a model that suitably fits experimental data, even in conditions different from those it was calibrated in, the model was used to explore five aspects of FT iron sequestration dynamics. The first aspect explored is the effect of the pre-existing number of iron atoms per cage in FT. Second, simulations were run to show the effect of FT concentration and APC under different experimental conditions than were used in the calibration and validation steps. Third, simulations with varying subunit compositions were run, driven by a curiosity about the consequences on iron sequestration at different heavy to light ferritin subunit ratios. The fourth aspect addressed is the dynamics of iron release from FT. Lastly, to demonstrate how this model can be used as a component of larger models, it is incorporated into a previously published model of hepatocyte iron metabolism [33], and it is shown how that model's properties improve when doing so.

### Effect of pre-existing iron core on FT iron sequestration dynamics

The model was used to simulate the effect that different initial values of iron atoms per ferritin core (APC) have on FT iron sequestration dynamics. The model was initialized using the conditions of the Macara *et al.* 1972 [31] dataset, namely the initial ferritin concentration was $2.32 \times 10^{-6}$M and LIP was set to $1.09 \times 10^{-3}$ M, corresponding to 470 iron atoms per cage. Simulations revealed a dependency of the mineralization rate over time on the initial APC values (Fig 3, time 0-15s). Generally, the curves showing mineralization rate fall into three groups dependent on initial APC. In the first group (<1000 APC), the mineralization rate starts low and then increases as iron accumulates inside FT, then later falls as the iron concentration in solution diminishes. For this group, Fig 3D shows the time it takes to get to the maximum mineralization rate decreases as the initial iron atoms per cage increases. In the second group (1000–3000 APC) the mineralization rate starts high and then falls quickly as all the iron is sequestered out of solution. In the third group (>3000 APC) the mineralization rate falls as the simulation proceeds, as in this case the accumulation of core iron slows the rate of new mineralization. Importantly, Fig 3B shows that although the mineralization rate varies in the first 20 seconds, the total FT iron accumulation at the end of simulation is about the same (FT core

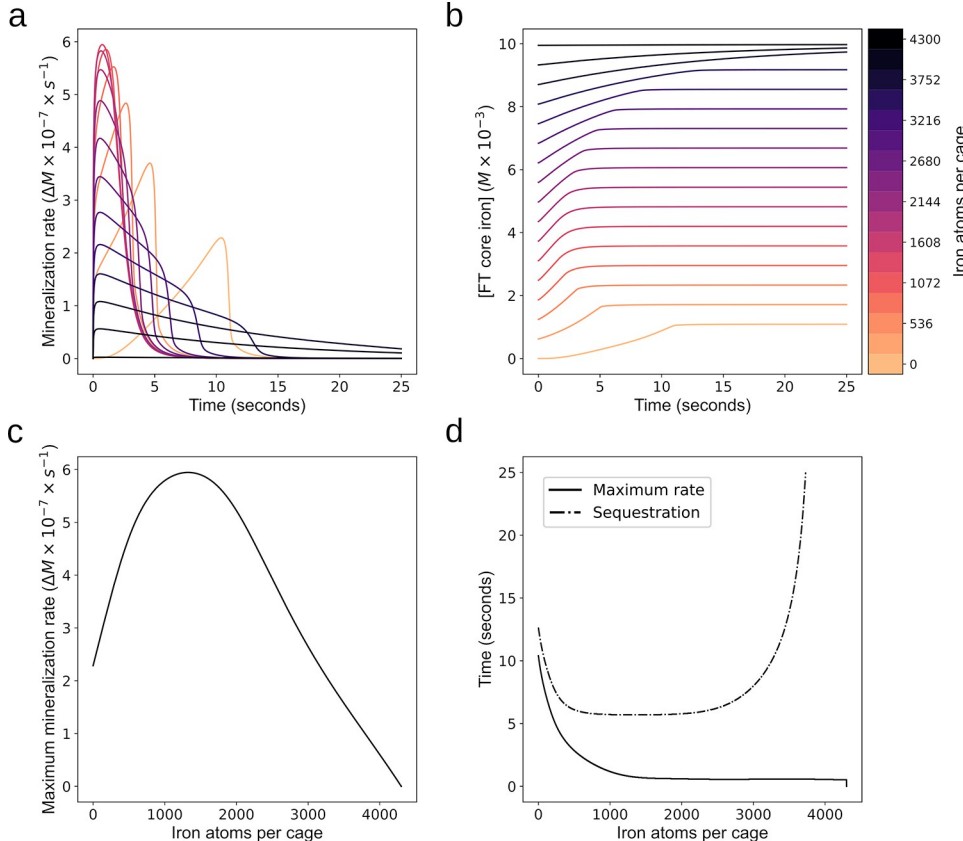

**Fig 3. Iron atom per core effect on FT dynamics. a. Mineralization rate over time as a function of iron APC. a** series of simulations were run for which the initial ferritin concentration was $2.32\times10^{-6}$M and iron added was $1.09\times10^{-3}$ M, corresponding to 470 iron atoms per cage, while the initial iron content (APC) varied from 0 to 4300 (as indicated by the colorbar on the right). **b. Core accumulation over time as a function of initial APC**. From the same simulations that generated the plots in A, here the total core mineralization is plotted over time with the same variation in iron per cage of 0 to 4300 atoms. **c. Maximum mineralization rate.** Plotted is the maximum value for the mineralization rate as a function of iron atoms per cage. This is the maximum value from the curve in **a** over the entire time course. **d. Time to sequestration and time to maximum mineralization rate**. Plotted is the time taken to reach the maximum value for the mineralization rate shown in c. Also plotted is the time taken for 99% of the LIP in solution to be sequestered dependent on initial iron atoms per cage. For values over 3729 atoms per cage it took longer than the 25 second time course.

iron increased by $1.09\times10^{-3}$ M, irrespective of initial amount of core). The difference in the final amount of FT core iron in Fig 3B is due to the different amount of initial core iron. Only in the cases where the APC values approach the limit of 4300 atoms per cage (beginning at >3500 APC) is the total FT core iron accumulation lower. These results suggest that APC plays a role in short term iron buffering (seconds) without significantly affecting the long term iron sequestration.

## Ferritin iron sequestration dynamics at the cellular scale

The model was calibrated against *in vitro* experimental data, where the concentrations of and iron are typically orders of magnitude larger than what is found in mammalian cells. Ultimately, the model was built with the purpose of being used for cellular models, so it was necessary to explore its behavior in the range of concentrations typically found in mammalian cells. Thus initial conditions of the model were modified so that the volume was set to $1.4 \times10^{-12}$ l,

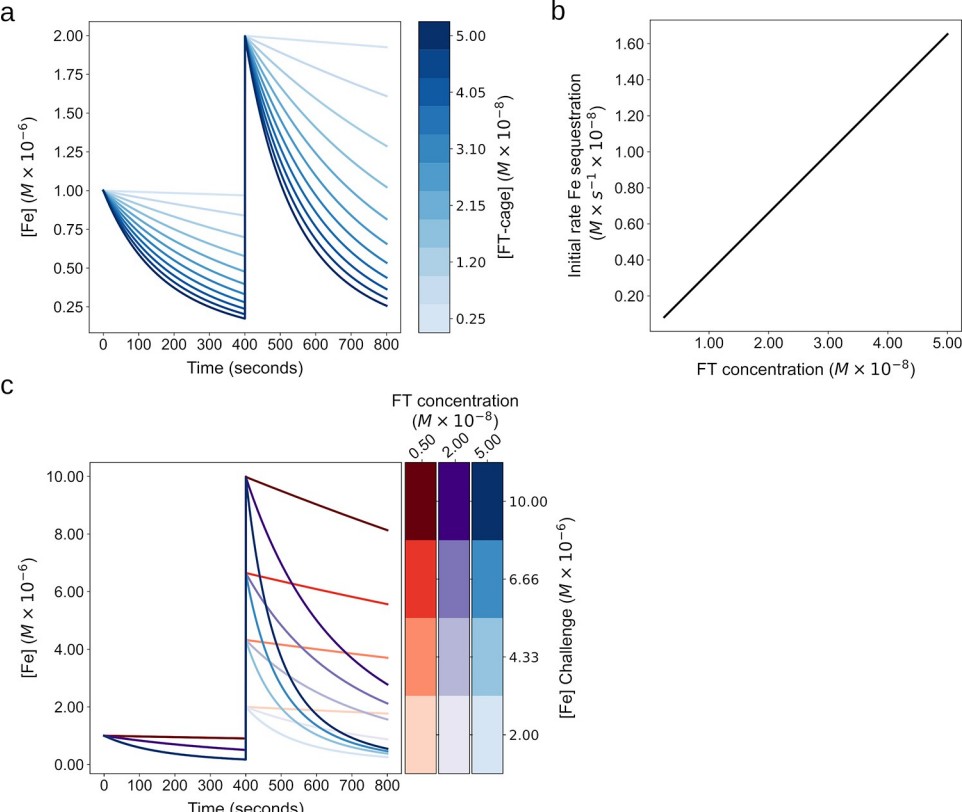

**Fig 4. Model simulation at the cell scale and analysis of FT concentration and iron per core in 2 factor tuning of FT buffering behavior. a. Effect of FT concentration on buffering behavior.** A series of simulations were run to determine the effect of FT concentration on the iron sequestration dynamics. Iron is added to the cytoplasm by setting LIP to 1μM at the simulation start and then set again at 400 seconds to 2μM. The curve shows the LIP remaining in solution over time in the presence of different FT concentrations (colorscale). **b. Effect of FT concentration on initial LIP sequestration rate.** The initial LIP sequestration rate across the same range of FT concentrations as in A is calculated following the addition of 1μM LIP. **c. FT buffering of large LIP additions.** A series of simulations were run to determine the LIP sequestration of larger LIP additions. Each curve indicates a simulation run at a FT concentration (5nM, 20nM, 50nM) and LIP concentration (2μM, 4.33μM, 6.66μM, 10μM).

LIP concentration was set to 1 μM and FT concentration to 5 nM. These values fall within a biologically plausible ranges [14, 50–55]. All reaction parameter values were kept identical to those in Table 1.

First analyzed was the effect of varying the FT concentration on LIP size in the above initial conditions and also after a modest LIP increase (Fig 4A). Simulation shows the slope at which iron is sequestered by FT is determined by the FT concentration with an increased slope resulting from increased FT concentration. The increase in LIP leads to increased iron sequestration by FT, with a rate directly proportional to FT concentration (Fig 4b).

What happens when there are large increases in LIP? Fig 4C shows simulations where LIP was increased in the presence of different concentrations of FT. The increase in LIP leads to increased iron sequestration by FT, again with a rate directly proportional to FT concentration.

Next the outcome of depleting the cytoplasmic LIP on iron release from FT was examined. Fig 5 shows that iron is released from FT immediately after having depleted LIP, and the effect of three FT concentrations. FT releases non-mineralized iron (from DFP) partially restoring

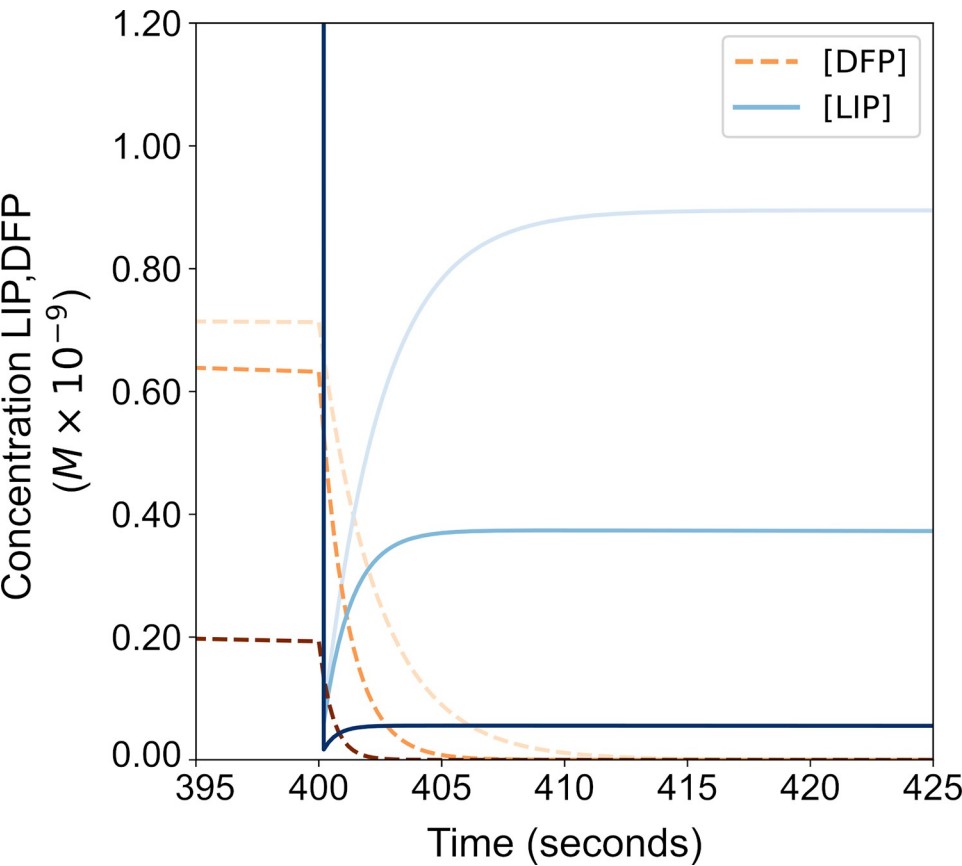

**Fig 5. Iron release from FT after LIP reduction.** At time 400 second the LIP was eliminated (reduced to 0 M) and the resulting DFP and LIP concentrations are plotted as iron is liberated from FT. Each curve indicates a simulation run at a FT concentration (5nM, 20nM, 50nM) (darker color is a higher FT concentration).

the cytoplasmic LIP. This is in agreement with the results of a prior model by Salgado *et al.* [32] which also suggested that it is only the non-mineralized iron that is available for immediate cytoplasmic iron replenishment.

Since the APC has shown to be influential *in vitro* (Fig 3), we next tested its effect in cellular conditions. As mentioned above, it has been shown that the APC alters the iron mineralization rate [40]. Simulation of a range of APC values between 0–4300 in cellular conditions shows that the APC modifies the slope of iron sequestration, but to a lesser degree than the FT concentration (Fig 6A and 6B). When both FT concentration and APC vary simultaneously, they work together to provide both a coarse (by FT concentration) and fine (by APC) tuning of iron buffering behavior (Fig 6C and 6D).

### Subunit composition

Unlike previous modeling attempts the present model takes into account the effect of the different FT subunits on iron sequestration. Noted previously, and reflected in the rate laws of this model, the H subunit catalyzes oxidation and the L subunit increases nucleation rate. We ran simulations with different H:L compositions, going from an L-homopolymer (0:24) to a H-homopolymer (24:0) to investigate how the model predicts the effect of H:L composition. Enhanced oxidation and mineralization is seen in polymers with greater H enrichment (Fig 7A), corroborating the

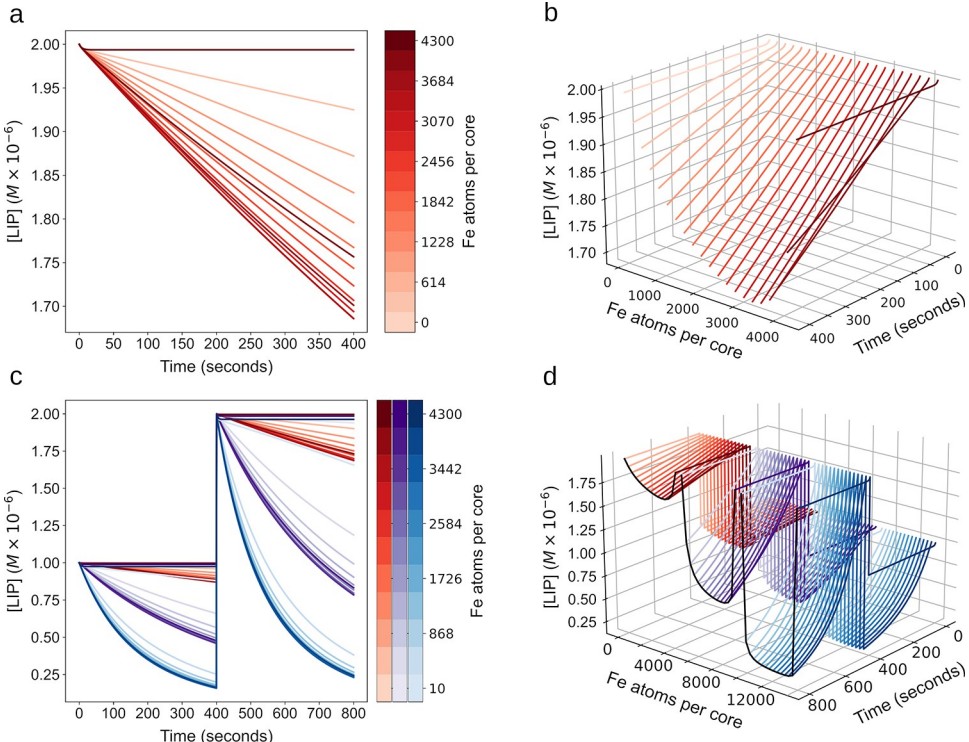

**Fig 6. Model simulation at the cell scale and analysis of FT concentration and iron per core in 2 factor tuning of FT buffering behavior.** Effect of iron per cage on buffering behavior. A series of simulations were run to determine the effect of varying the initial core size (APC) so that the cages at an identical FT concentration ($5 \times 10^{-9}$ M) had different iron content from 0 to 4300 APC. **a** shows the curves in 2D with LIP remaining after a 2.5 µM LIP addition at time 0; in **b** the APC is added as a 3rd dimension for visual clarity. **Simultaneous variation of FT concentration and APC on buffering behavior.** A series of runs at 3 FT concentrations were run $5 \times 10^{-9}$ M (red), $2 \times 10^{-8}$ M (purple), $5 \times 10^{-8}$ M (blue). For **c** and **d** each run in addition to the FT concentration, the APC was varied between 0 and 4300 as well.

finding of Mehlenbacher et al. 2017 [42] that "at a high iron flux, the few H-subunits present in L-rich ferritin cannot sustain the oxidation of Fe(II) at the ferroxidase sites and the protein favors iron oxidation on the surface of a growing mineral core" and Fig 8 in that article. Interestingly, though a larger number of L subunits should better activate the nucleation rate according to the rate law used in the model (Eq 3), simulations show that the model of L-homopolymer has a lower observed nucleation rate than any of the other compositions (Fig 7C) and has a reduced LIP sequestration rate compared to cages containing H subunits (Fig 7A). This result suggests that, in the present model, nucleation depends more on the presence of DFP produced through H-catalyzed oxidation than on the presence of L-catalyzed nucleation rate. In the case of the L-homopolymer the effective nucleation rate can be seen to fall as the simulation proceeds. This may be because while L enhances nucleation, once a single crystal has begun to form, DFP no longer requires additional nucleation steps for continued mineralization. Indeed, electron micrographs of FT cages often show one to a few crystals present within a FT cage, indicating that nucleation may be a relatively rare event [46]. Obviously these are model predictions that depend on the values of the rO and rN phenomenological parameters and it may well be that the L subunit could have a stronger effect on nucleation, though this is not clear from the literature. These results indicate that the effect of the L subunit on nucleation versus the rate at which DFP is added to a single crystal should be studied, allowing for better values of the rN model parameter, as this would better explain the different effects of the H:L composition of ferritin. In summary,

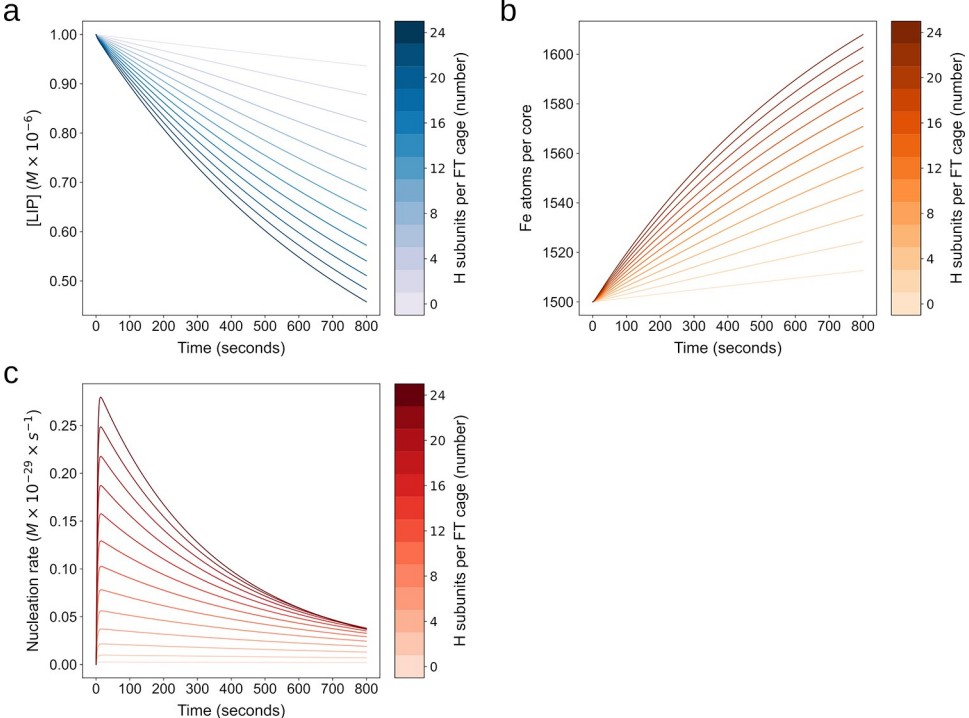

**Fig 7. Subunit composition effect on FT iron sequestration dynamics.** The model contains a parameter H that can be varied to incorporate differences in subunit composition of a population of FT-cages versus another and the effect of the composition on the rate laws of oxidation and nucleation. H represents the number of H subunits and can be between 0 (L homopolymer) and 24 (H homopolymer). We ran 13 simulations with the number of H subunits in the FT cages increasing from 0 to 24 in increments of 2. Then iron sequestration behavior was assessed by tracking **a)** LIP **b)** APC and **c)** nucleation rate over time.

though the details require further experimental characterization, subunit composition may be a third mechanism by which iron buffering by ferritin can be modulated. A critical distinction is that when varying subunit concentration, the steady state LIP reached will be the same independently of composition, it is just the time to reach that steady state that is affected by the FT subunit composition.

## Iron release from FT

The question of whether or not there is physiologic release of iron from FT, without lysis of FT cages, to replenish cytosolic stores when LIP falls is still under investigation [25, 56]. It has been shown that such release is possible through the action of iron chelators and reducing agents [57–61], but it is not clear whether it happens without these agents. A different mechanism for release of iron from FT is through destruction of the ferritin cages through autophagy [25, 56, 62] and NCOA4 regulated ferritinophagy has been well characterized [63]. Even though it is still unclear whether iron release from FT without FT degradation happens physiologically, it is possible to include such a mechanism in the model as a hypothesis and examine its consequence, which we do here.

To incorporate iron release from the core, we adapted a proposed mechanism from the hepatocyte model of Mitchell et al. [33]. In this scheme, FT can be synthesized by a reaction representing protein synthesis and degraded by another reaction (both with mass action kinetics). In line with this, when FT is degraded the proportion of total core present within the

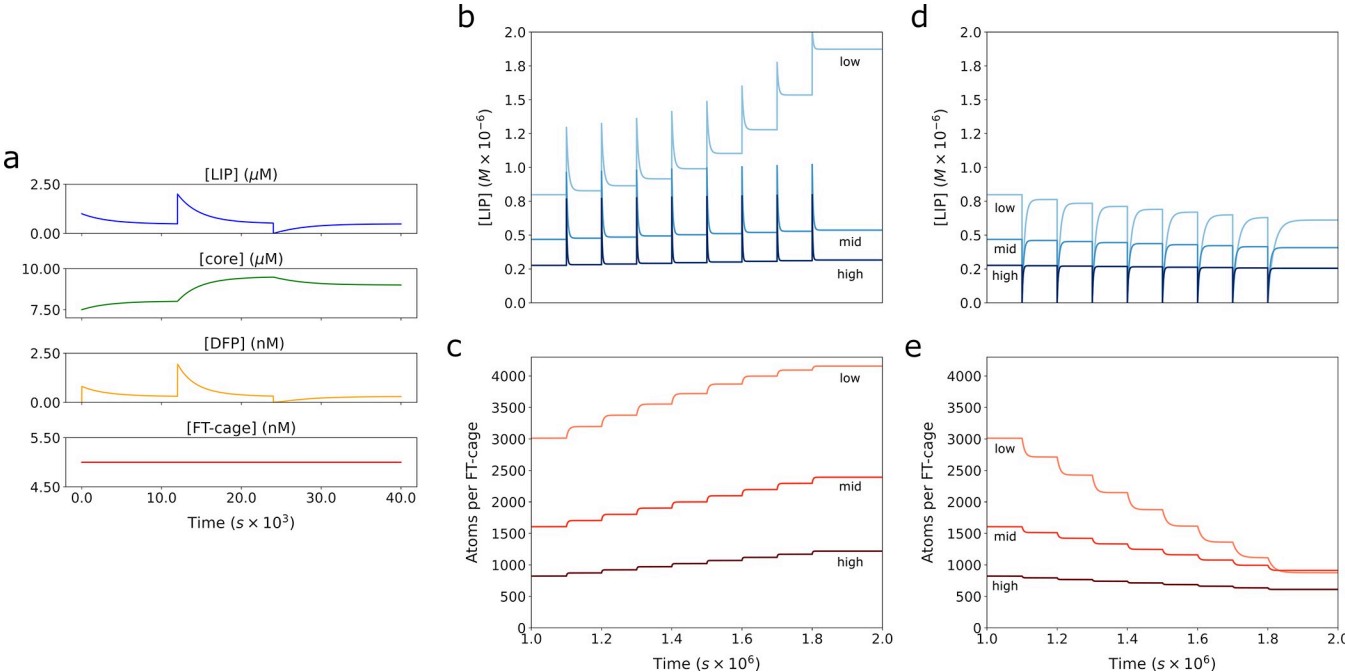

**Fig 8. Model dynamics with FT core iron release.** Release was incorporated into the model and balanced by expression as noted above to reach an equilibrium FT concentration of 5nM. The model was then run for 10,000 seconds to let the system approach steady state (where LIP did not change over time). Then LIP was doubled and subsequently halved (at time 25,000). **a.** The amount of LIP (blue), core (green), DFP (orange) and FT-cage (red) concentrations are plotted over time. For analyzing behavior over the long term and response to repeated iron challenges (**b**—**e**), three increasing expression rate values (3.075x10⁻¹⁴, 6.015x10⁻¹⁴, 1.203x10⁻¹³) were chosen for the long term sequestration analysis ("low", "mid", "high" curve labels respectively). These rates corresponded to equilibrium concentrations of FT of 2.56nM, 5nM, and 10nM. For simulation every $10^5$ s (about 1 day) the LIP was increased by 0.5μM, a small physiologic dose. The **b**) LIP and **c**) APC were plotted corresponding to the model with each of these expression rate values over time. The reverse was also performed, where every $10^5$ s the LIP was reduced to 0 and again the **d**) LIP and **e**) APC were plotted.

amount of FT degraded is released into the LIP. We used the 1.203x10⁻⁵ s⁻¹ degradation rate constant value from the Mitchell model directly (the value of which was originally calculated from data in Kidane et al., 2006 [25]) and set the expression rate constant to 6x10⁻¹⁴ s⁻¹. This parameterization results in FT reaching an equilibrium concentration of 5nM (Figs 5–7). Next we ran simulations to test how much the model behavior changed with these updates. In Fig 8, a time course was run where the amount of LIP was doubled at a certain time, and subsequently zeroed at a later time. The changes in LIP and the amount of iron present within FT was tracked (Fig 8A). The doubling of LIP size is quickly suppressed (LIP returns to near its initial concentration) showing the buffering value of FT; the depletion of LIP was also buffered, with iron quickly released from FT to restore the LIP to its steady state value. These results are qualitatively similar to those of Salgado et al. 2010 [32].

Next we tested the model behavior at longer timescales (hours to days). Because of the intended use in cell scale models, the expression rate constant was set to three sequentially increasing values to mimic cells maintaining different levels of FT expression. This has the effect of establishing 3 steady state concentrations for FT and LIP. Once the steady state is achieved, at regular timepoints, once per day (~$10^5$ s), the LIP is increased to simulate repeated iron loads (Fig 8B and 8C). Note that the expression and degradation rate stayed the same so the total FT concentration did not change. The opposite perturbation was also carried out on these three FT concentration levels, where LIP was reduced to 0 (Fig 8D and 8E). These simulations indicate that higher FT concentrations are better able to buffer repeated iron loads and

maintain LIP at a steady concentration. Lower concentrations of FT result in increasing steady state LIP (when adding iron) or in decreasing steady state LIP (when removing iron).

## ModelBricks application use case

A major aim of the present model is to be used as a sub-model of FT iron storage for incorporation into larger cellular models of iron metabolism. As a proof of principle for this use we now apply it to a previously published model of cellular iron metabolism, the hepatocyte model of Mitchell et al. 2013 [33]. In this model, there is a hepatocyte cell with a FT concentration that internalizes LIP, in addition to several other iron related processes. However, in contrast with the model developed here, the model of Mitchell et al. makes no distinction of the state of the iron within the FT, the ferroxidase activity and DFP formation are also not mechanistically represented there, and the process of mineralization is represented by mass action kinetics. Thus we substituted the FT-related reactions of that model with the model developed here, thus using it as a model brick. Additionally, we were also able to include subunit detail with the model developed here, and set the "H" parameter to 4 to reflect what is known about human hepatocyte FT subunit composition [64]. We then analyze the outputs of the original model of Mitchell *et al.* and the new version incorporating the present FT model (BioModels MODEL2211030002). The main results are depicted in Fig 9. For simulation, all parameters of both FT representations are kept the same, except for the amount of FT in the core. The core difference is because the previous model FT representation assumes optimum FT mineralization rates at all FT iron content values and so to match this initial condition we set the APC in the new version to 1500, which best matches that optimal rate. For model simulation, to allow for direct comparison irrespective of which FT submodel is included, at time 0, the [FT] was 1.66 nM and LIP was 1.3 μM. Tracked was the FT-associated iron, represented differently in each, but representing the same FT core concept.

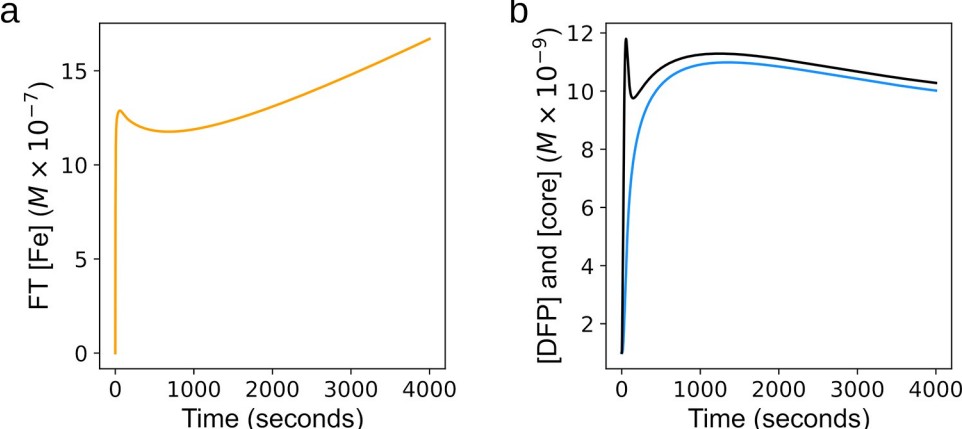

**Fig 9. Model comparison.** To analyze the effect of substituting the model presented in this work with the FT representation of a previous model, a simulation from the previous model was run and the behavior of FT associated iron compared. For simulation, all parameters of both models are kept the same, except for the amount of FT in the core (due to the new model's dependence of mineralization on APC). At time 0, the FT concentration was 1.66 nM and LIP was increased to 1.3 μM. Tracked was the newly accumulated FT associated iron after the LIP increase, represented differently in each, but representing the same FT core. **a.** FT iron (orange) accumulation over time in original hepatocyte model from Mitchell et al. 2013 [33] **b.** New FT core (blue) and core plus DFP (black) accumulation over time in hepatocyte with new FT model. Final FT concentrations were 37.5nM and 15.8nM respectively.

Comparing the two model outputs (the original Mitchell et al. model and the one with this FT submodel included) we observe three main points. The first is that the behavior of the model with the FT submodel presented here is qualitatively similar to output in a very general sense when the previous FT submodel is maintained. With both submodels there is a sharp peak in FT iron early in the simulation, followed by decrease in FT iron, and then followed by a gentler increasing slope. The second is that the new submodel enables a deeper understanding of the FT mineralization. Due to the new submodel's increased detail, it can be seen that the peak is actually a result of DFP concentration rising and not the mineralized core. This realization is critical because DFP seems to be more readily released back into the cytoplasmic LIP than the iron in the mineralized core, so if there was a temporally relevant event affecting LIP after this DFP formation, our submodel would be expected to buffer any LIP change initially through DFP. Lastly we note that the curves have different behavior both quantitatively throughout the time course and particularly at steady state. The submodels produce model output that is most different after 1000 seconds in the simulation. With the original FT representation there is a gradual increase in FT iron while with the new submodel a decrease in FT iron is observed. A possible explanation for this behavior is the FT kinetics of the new submodel result in the loss of hepatocyte LIP being buffered by FT (by FT iron release) whereas in the original model other cellular mechanisms in the model dominated the changes in LIP. This general qualitative similarity but quantitative and steady state differences help to further solidify the utility of the submodel presented here.

## Discussion

The model of ferritin iron sequestration and release presented here was constructed, calibrated, and validated using experimental data from several independent studies and laboratories. This model's behavior is consistent with the conclusions of the previous models that "the main function of ferritin is to operate as an extremely fast, high-capacity, iron buffer that maintains mobile iron concentrations (cLIP)" [32]. We believe the present model is available in a suitable standard format and has an appropriate level of detail for incorporation as a component into larger cellular models, while still capturing essential aspects of the dynamics of FT iron sequestration. Our hope is that it can serve as a base for other modelers and reduce the time and effort needed for FT representation within new models. Modelers can use it as is, but depending on the specific purpose for which new models are constructed it may need to be tuned or expanded. To make it more specific to a particular organism or cell type, some parameters may need to be replaced by others more representative of that system. We present an example of using it as is by incorporating this FT template model into a larger hepatocyte model (by substituting in our model for its more simplistic representation of FT iron sequestration). An example of tuning this model for a specialized case would be to represent mitochondrial ferritin (MtF). MtF is a homopolymer of 24 subunits that have a high degree of sequence homology with human H-chain ferritin (HuHF) [65, 66]. Thus, the model with the H parameter set to 24 could serve as a starting point, but it may be better to adjust the parameter values specifically to MtF data for a more accurate model. Indeed, MtF and HuHF have been shown to have striking differences in their ferroxidase activity, despite their high level of homology [65]. A model expansion example case would be to consider the interaction of other metals that can also be incorporated into the ferritin core. To do so, new species and reactions would need to be added, including the presence of other metals together with appropriate reactions for their incorporation into the core and interactions with iron. A more detailed description of how to add/modify this FT submodel to an existing or future model is available in the Supporting information.

Beyond its purpose of being a submodel (a "ModelBrick"), the present model is also used to explore general properties of FT iron sequestration behavior. Model simulations show how FT concentration, the amount of iron present in each cage, and subunit composition can tune the iron sequestration rate of FT, ultimately influencing transient and steady state levels of cytoplasmic label iron pool. In addition to accurately reproducing previous experimental studies, the performance of the model provides novel hypotheses regarding the FT iron sequestration behavior.

The simulations run to determine the effect of pre-existing APC on iron sequestration suggest important experimental implications. When interpreting the results of Fig 3, the variation in mineralization rate at APC values of 0–100 suggests that even a small amount of iron retained within the FT cages can influence the measured mineralization curve, making the control of experimental error and limiting of incomplete iron depletion from FT necessary for proper interpretation. The influence of a few residual iron atoms not accounted for when running the model is one possible explanation for why in Fig 2C the validation steady state seems to be a good fit but the initial part of the simulated time course deviates from the experimental observations; the model assumes 0 initial iron inside FT but if that is not the case in the experimental setting the model would not be accurate (i.e. even if only very low amounts of sequestered iron are present experimentally). In fact, when the APC is increased to between 75–100 in the model, the fit to the data is improved. Outside of the model congruence with experimental data, the APC effect and the lack of experiments where APC is larger than a few atoms suggests the effect of this variable should be studied experimentally much more in depth. Most experiments are set up using apoferritin (ferritin with no internal iron), which is either reconstituted or generated from cellular ferritin and iron is depleted using reducing agents/iron chelators [59–61]. These conditions are good for determining initial rates of iron incorporation, but physiologically the distribution of iron atoms per FT cage is rarely that few [32].

It has been observed that ferric iron can move out of FT with the aid of reducing agents or chelators [59–61]. Alternatively it is well known that iron can be released from FT by degradation of the FT within lysosomes [25, 56, 62]. What is not clear is whether there is significant release of iron from FT cages *in vivo* without lysis of FT, though that is being investigated experimentally [21, 23, 39]. Particularly exciting is the discovery of the role of NCOA4 in recruiting ferritin into lysosomes, eventually allowing the stored iron to be released. While NCOA4-driven release is not coded explicitly in the present model, it does include a release mechanism representing FT degradation and consequently release of its core iron. The rate law used can describe multiple mechanisms and has the flexibility to be adapted in the future to include additional modifiers (eg NCOA4) while preserving the rest of the model components.

Lastly, the simulations of steady state and long term time courses of LIP and the corresponding APC (Fig 8) indicate that by altering only the FT concentration a cell would have sufficient control to tune iron buffering and therefore reach a desired steady state cytosolic iron level. This conclusion arises from an analysis of the simulations in Fig 8. In Fig 8B, before the first iron dose, the three different FT concentrations result in different LIP levels. This LIP variation reinforces the idea that the steady-state LIP level depends on the FT concentration, presented first by Fig 4 simulations. Then the simulations in Fig 8B and 8D indicate that in the presence of repeated iron dosing, higher FT concentrations manage to keep the LIP at its initial steady state level, while the lower FT concentration eventually loses its buffering ability. This loss is because at the lower concentration, all FT cages fill up with iron and eventually can no longer accept more iron (as evidenced by the APC curve approaching 4300 in Fig 8C). This result suggests that in addition to transiently increasing LIP by liberating iron from FT, a cell's steady state cytoplasmic labile iron concentration could be increased by lowering FT

concentration, either by reduced expression rate or increased degradation rate. Further, over time with accumulated iron doses the lower FT concentration also leads to higher APC making the FT cages that are present less able to buffer incoming iron (Fig 8C). In an iron deprived state, this decreased buffering would allow iron to be used for other processes rather than sequestered by FT. If the APC rises to a point where the buffering effect disappears and LIP rises, a modest increase in FT driven by increased expression could restore that lost buffering capacity. The relevance of these results is highlighted by a recent publication that showed the importance of cellular FT concentration: in humans cells have a much lower tolerance for noise fluctuations in FT concentration than other proteins [67].

## Supporting information

**S1 File. Full set of model equations.** File contains complete set of model equations. (PDF)

**S2 File. Model files and model modification instructions zip.** Zip file containing all model files and considerations for adding the FT sub model to a cell model. (ZIP)

**S3 File. Experimental data files zip.** Zip file containing files for all experimental data used in calibration and validation of the model. (ZIP)

## Acknowledgments

We thank Rudradeep Mukherjee and Drs. A. Cowan, M. Blinov, S. Torti, and P. Vera-Licona for discussions about this project.

## Author Contributions

**Conceptualization:** Joseph Masison, Pedro Mendes.

**Data curation:** Joseph Masison.

**Formal analysis:** Joseph Masison, Pedro Mendes.

**Funding acquisition:** Pedro Mendes.

**Investigation:** Joseph Masison.

**Methodology:** Joseph Masison.

**Project administration:** Pedro Mendes.

**Resources:** Pedro Mendes.

**Software:** Joseph Masison.

**Supervision:** Pedro Mendes.

**Validation:** Joseph Masison.

**Writing – original draft:** Joseph Masison.

**Writing – review & editing:** Joseph Masison, Pedro Mendes.

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
