## [Decision Letter · Decision Letter 0]

4 Jan 2023

PONE-D-22-31702Modeling the iron storage protein ferritin reveals how residual ferrihydrite iron determines initial ferritin iron sequestration kineticsPLOS ONE

Dear Dr. Mendes,

Thank you for submitting your manuscript to PLOS ONE. After careful consideration, we feel that it has merit but does not fully meet PLOS ONE’s publication criteria as it currently stands. Therefore, we invite you to submit a revised version of the manuscript that addresses the points raised during the review process.

We look forward to receiving your revised manuscript.

Kind regards,

Shailender Kumar Verma, Ph.D.

Academic Editor

PLOS ONE

Journal Requirements:

"We thank Rudradeep Mukherjee and Drs. A. Cowan, M. Blinov, S. Torti, and P. Vera-Licona for discussions about this project. P. M. is funded by the NIH grant R24 GM137787, National Resource for Mechanistic Modeling of Cellular Systems."

Reviewers' comments:

Reviewer's Responses to Questions

**Comments to the Author**

1. Is the manuscript technically sound, and do the data support the conclusions?

Reviewer #1: Yes

Reviewer #2: Yes

Reviewer #3: No

2. Has the statistical analysis been performed appropriately and rigorously? 

Reviewer #1: Yes

Reviewer #2: Yes

Reviewer #3: No

3. Have the authors made all data underlying the findings in their manuscript fully available?

Reviewer #1: Yes

Reviewer #2: Yes

Reviewer #3: No

4. Is the manuscript presented in an intelligible fashion and written in standard English?

Reviewer #1: Yes

Reviewer #2: Yes

Reviewer #3: Yes

5. Review Comments to the Author

Reviewer #1: In the manuscript titled with “modeling the iron storage protein ferritin reveals how residual ferrihydrite iron determines initial ferritin iron sequestration kinetics”, the authors described an important issue with regarding to modelling the iron sequestration kinetics by ferritin. In comparison to previous efforts, they have simplified the process into 4 basic steps and included addition parameters such as subunits compositions, APC etc to accurately describe this process. Finally, the authors try to connect this to large iron metabolism in different cell types. Overall, this manuscript proposed a good model to plug in ferritin part of iron metabolism, however, there are some places the authors could use some improvements.

1. The abstract is quite disconnected from the introduction. The abstract majorly focused on the iron metabolism and the intro starts with the modelling. There should be a better way to present this work to make its flow more consistent.

2. In figure 2 which is the calibration and validation process, the predictions seem quite different from the real data. Could the authors add some statistical measurement to show how close their prediction is to the real data?

3. The authors stated that including the subunit composition is one of the highlights for this paper. So is there any situations the cells or human adjust the ratio of H-subunit to L-subunit to change the ferritin composition? And can your data match those conditions?

Reviewer #2: In this manuscript entitled “Modeling the iron storage protein ferritin reveals how residual ferrihydrite iron determines initial ferritin iron sequestration kinetics” by Masison & Mendes, the authors describe a new model that describes the sequestration of iron by the iron storage protein ferritin. Ferritin is a crucial and very complex ferritin and to-date we still understand very little as to how iron is sequestered and stored by this multimer-forming biopolymer.

The proposed model here is interesting and sheds light on a rather complex biochemical mechanism. This will help the field greatly. Interestingly, the authors relate the results of their model with experimental data from other studies, which gives more confidence in this model.

However, there are a few points the authors should address, before this work can be recommended for publication.

Points to address:

1) There are some reports that copper and other metals can also be stored by ferritin. What is the take on that by the authors? Since iron sequestration determines the sequestration kinetics, how do other cations influence the kinetics? Could the authors model with concentrations and crowding present in cells, or at least comment and discuss this?

2) A recent review greatly summarized the chemistry and biology of ferritin, including the formation of the polymers and discussing the different models of iron sequestration known. Please cite it here (Metallomics. 2021 May 12;13(5):mfab021. doi: 10.1093/mtomcs/mfab021).

3) FT can have different ratios of FTH and FTL. How does this influence the proposed model by the authors?

4) Can the authors comment if their model can be transferred to mitochondrial ferritin? This should be at least discussed.

Reviewer #3: The authors present a model that attempts to describe the kinetics of iron mineralization in ferritin by incorporating the essential features of Fe(II) transport, binding, oxidation, nucleation and mineralization. Unfortunately, the model neglects several features that have been identified in experiments, including 3-site binding at the ferroxidase site, with binding at the third site helping to dislodge the di-iron DFP from the ferroxidase site; the option for iron oxidation and subsequent mineralization occurring directly at the mineral surface; the transient role of the diferric peroxo intermediate, the role of the mu-oxo diferric intermediate, and the role of water deprotonation in the mineralization process. In the common H:L ferritin heteropolymers, it makes sense that most or all L ferritins could be engaged in nucleating mineral formation, which would help to spread the mineral more broadly over the protein surface and allow for more complete filling of the protein cavity. The authors instead advance the hypothesis that only a small number of nucleation sites are involved, without providing evidence.

The authors' models do not agree particularly well with the experimental data in figure 2, in particular Figs 2C and 2E. Little justification is given for these discrepancies, but they imply deficiencies in the model.

If understanding biology is the goal, it seems more realistic to consider mineralization kinetics under conditions of low levels of free iron. Is the addition of thousands of equivalents of Fe(II) per ferritin 24mer relevant to physiologic conditions?

In the physiologic scenario, there is presumably rarely an empty "apoferritin", and most mineralization must occur at partially filled ferritin cavities. Understanding the details of how partially filled ferritin mineralizes additional free iron is important to model accurately.

The authors only consider the diferricperoxo species, even though this is short-lived and is known to decay to diferric oxo or hydroxo species.

There is strong experimental evidence that under low- or high-iron-loading conditions in ferritin heteropolymers, the ferroxidase centers control chemical reactivity and generate DFP species only at initial stages of mineralization. (J. Mol. Biol. (2005) 352, 467–477). At subsequent stages, mineralization can be modeled to occur mostly at the mineral core via an autocatalytic mechanism.

The authors do not appear to take the available experimental data for heteropolymers into account in developing their own model.

It may be too challenging, confusing and generally not useful to compare pure H homopolymer to H+L heteropolymers in this work, based on what are likely to be very different mineralization mechanisms. The authors unfortunately gloss over these differences despite considerable work by Chasteen, Theil and others comparing homo- vs. heteropolymer biomineralization.

In general, the authors appear to neglect the fact that several of the better ferritin manuscripts in the literature (incl the 2005 JMB paper cited above) provided a full kinetic analysis, and model the different species that are formed based on UV-Vis spectroscopy. These studies succeed in explaining mineralization kinetics for a specific ferritin heteropolymer, e.g., horse spleen apoferritin. The authors here are somewhat vague about which ferritin they are trying to model, but it seems to be overreaching to try to explain ALL ferritin biochemistry with this simplistic model. The DFP species may vary in its prevalence and mechanistic significance depending on the ferritin composition, etc.

6. PLOS authors have the option to publish the peer review history of their article (what does this mean?). If published, this will include your full peer review and any attached files.

Reviewer #1: No

Reviewer #2: No

Reviewer #3: No

---

## [Author Response · Author response to Decision Letter 0]

18 Jan 2023

Please see "Response to Reviewers" document which includes detailed responses.

---

## [Decision Letter · Decision Letter 1]

23 Jan 2023

Modeling the iron storage protein ferritin reveals how residual ferrihydrite iron determines initial ferritin iron sequestration kinetics

PONE-D-22-31702R1

Dear Dr. Mendes,

We’re pleased to inform you that your manuscript has been judged scientifically suitable for publication and will be formally accepted for publication once it meets all outstanding technical requirements.

Kind regards,

Shailender Kumar Verma, Ph.D.

Academic Editor

PLOS ONE

Additional Editor Comments (optional):

Reviewers' comments:

Reviewer's Responses to Questions

**Comments to the Author**

1. If the authors have adequately addressed your comments raised in a previous round of review and you feel that this manuscript is now acceptable for publication, you may indicate that here to bypass the “Comments to the Author” section, enter your conflict of interest statement in the “Confidential to Editor” section, and submit your "Accept" recommendation.

Reviewer #1: (No Response)

Reviewer #2: All comments have been addressed

Reviewer #3: All comments have been addressed

2. Is the manuscript technically sound, and do the data support the conclusions?

Reviewer #1: (No Response)

Reviewer #2: Yes

Reviewer #3: Yes

3. Has the statistical analysis been performed appropriately and rigorously? 

Reviewer #1: (No Response)

Reviewer #2: Yes

Reviewer #3: Yes

4. Have the authors made all data underlying the findings in their manuscript fully available?

Reviewer #1: (No Response)

Reviewer #2: Yes

Reviewer #3: Yes

5. Is the manuscript presented in an intelligible fashion and written in standard English?

Reviewer #1: (No Response)

Reviewer #2: Yes

Reviewer #3: Yes

6. Review Comments to the Author

Reviewer #1: I have read the response letter and the revised manuscript. In my opinion, the authors have well addressed most of the reviewers’ questions.

Reviewer #2: All the concerns of this reviewer have been addressed in this revised version.

There are no additional comments.

Reviewer #3: MS is greatly improved in revision. Authors did a nice job of addressing all critiques. Contributions made by this paper have been clarified in the text and improved in the figures.

7. PLOS authors have the option to publish the peer review history of their article (what does this mean?). If published, this will include your full peer review and any attached files.

Reviewer #1: **Yes: **Minmin Liang

Reviewer #2: No

Reviewer #3: No

---

## [Editor Report · Acceptance letter]

26 Jan 2023

PONE-D-22-31702R1 

Modeling the iron storage protein ferritin reveals how residual ferrihydrite iron determines initial ferritin iron sequestration kinetics 

Dear Dr. Mendes:

I'm pleased to inform you that your manuscript has been deemed suitable for publication in PLOS ONE. Congratulations! Your manuscript is now with our production department. 

Kind regards, 

on behalf of

Dr. Shailender Kumar Verma 

Academic Editor

PLOS ONE